# Progressive Prototype Evolving for Dual-Forgetting Mitigation in Non-Exemplar Online Continual Learning

## ABSTRACT

Online Continual Learning (OCL) aims at learning a model through a sequence of single-pass data, usually encountering the challenges of catastrophic forgetting both between different learning stages and within a stage. Currently, existing OCL methods address these issues by replaying part of previous data but inevitably raise data privacy concerns and stand in contrast to the setting of online learning where data can only be accessed once. Moreover, their performance will dramatically drop without any replay buffer. In this paper, we propose a Non-Exemplar Online Continual Learning method named Progressive Prototype Evolving (PPE). The core of our PPE is to progressively learn class-specific prototypes during the online learning phase without reusing any previously seen data. Meanwhile, the progressive prototypes of the current learning stage, serving as the accumulated knowledge of different classes, are fed back to the model to mitigate intra-stage forgetting. Additionally, to resist inter-stage forgetting, we introduce the Prototype Similarity Preserving and Prototype-Guided Gradient Constraint modules which distill and leverage the historical knowledge conveyed by prototypes to regularize the one-way model learning. Consequently, extensive experiments on three widely used datasets demonstrate the superiority of the proposed PPE against the state-of-the-art exemplar-based OCL approaches. Our code will be released.

## CCS CONCEPTS

• **Computing methodologies → Computer vision**.

## KEYWORDS

Continual Learning, Online Learning, Non-Exemplar

## 1 INTRODUCTION

In the domain of deep learning, the practical utilization of deep models has prompted a growing interest in continuous learning (CL) [49]. This popular paradigm has gained prominence due to its important role in addressing the dynamic nature of non-stationary data streams stemming from various downstream tasks [28, 35, 40]. The core of CL is to tackle the critical challenge of catastrophic forgetting which necessitates that a deep model strikes a delicate balance between efficiently accumulating new knowledge from new learning stages while preserving the historical knowledge acquired

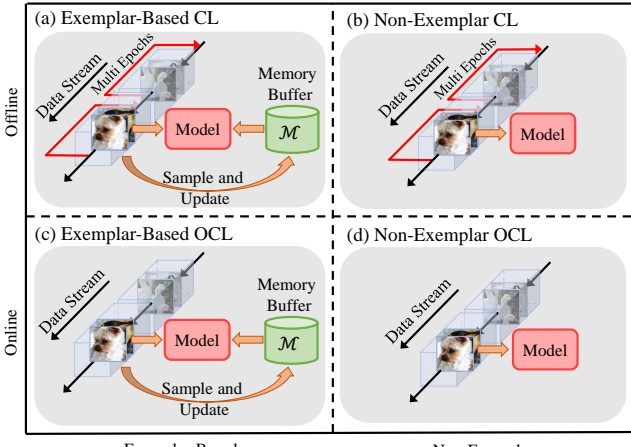

**Figure 1: The comparison between (a) the Exemplar-Based CL, (b) the Non-Exemplar CL, (c) the Exemplar-Based OCL, and (d) the investigated Non-Exemplar OCL in this paper. In the last scenario, all data can only be learned for one epoch and none of them can be retained as exemplars for reusing.**

from old stages [9]. Of particular note is that during a continuously non-stationary data stream, a unique dilemma emerges. Each sample can be accessed only once for learning which will further exacerbate the catastrophic forgetting problem. Consequently, there is a rising focus on a more practical but challenging CL scenario known as *Online Continual Learning* (OCL) [29] where each sample can only be accessed once, as depicted in Figure 1(c) and (d).

Due to the one-pass data stream in OCL, besides the catastrophic forgetting of knowledge between different learning stages, named *inter-stage forgetting*, deep models may also forget the previously learned knowledge within a learning stage and thus suffer severely from insufficient training. As depicted in Figure 2, the model may severely forget knowledge of data encountered earlier within a learning stage, and such a phenomenon is named *intra-stage forgetting* in this paper. As a result, the performance of existing CL methods is greatly limited in this challenging scenario. Recently, various OCL methods [1, 2, 20, 41] follow the same manner as previous exemplar-based CL approaches to store exemplars from previous learning stages for rehearsal. Nevertheless, apart from raising critical concerns about data privacy, these approaches stand in contrast to the essence of the online setting where each sample can only be accessed once.

Therefore, *how to effectively address OCL without using any previous exemplars* still remains challenging and unsolved, named as Non-Exemplar Online Continual Learning (NEOCL), as shown in Figure 1(d). Indeed, there are rare works focusing on this challenging NEOCL problem. [14] introduced a gradient projection

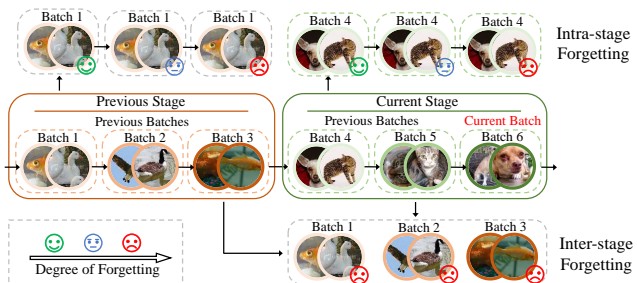

**Figure 2: The illustration of the intra-stage forgetting and inter-stage forgetting in OCL. Intra-stage forgetting means the model forgets the knowledge of data encountered earlier within a training stage. Inter-stage forgetting refers to the model's tendency to forget classes from previous stages.**

strategy to constrain the updating of model parameters during OCL. The latest work [19] pre-trained an offline model on half of the datasets and subsequently froze the backbone during the online learning phase. While these strict constraints on model updating alleviate the catastrophic forgetting issue to some extent, they also impose significant limitations on the learning capacity. This restriction becomes particularly apparent when a substantial portion of the pre-training data is inaccessible. Recently, to resist forgetting without exemplars, various prototype-based methods have been investigated in Non-Exemplar Continual Learning (NECL) [52, 53], which compute the mean feature of all samples from the same class after a learning stage, serving as the prototypes. However, these approaches only tackle inter-stage forgetting and are infeasible in NEOCL due to the single-pass data stream.

To handle both the intra and inter-stage forgetting issues, we propose a novel NEOCL method called Progressive Prototype Evolution (PPE). Our PPE approach leverages learnable class-specific prototypes for each class as surrogates of previously acquired knowledge to facilitate the learning of OCL. Specifically, the prototypes, treated as learnable parameters, are progressively learned and evolved along with the OCL process to convey more informative knowledge. By involving them in the learning of the current OCL stage through a prototype feedback design, the intra-stage forgetting issue can be greatly alleviated. Additionally, to tackle inter-stage forgetting, two vital components, the Prototype Similarity Preserving and Prototype-Guided Gradient Constraint modules, are proposed based on the obtained prototypes. The former aims to distill the similarity knowledge between the learned prototypes and features of input data, regularizing the model and resisting forgetting from the data perspective. Furthermore, the inter-stage forgetting caused by model updating will inevitably deteriorate the representation ability of the prototypes of previous stages. Therefore, the Prototype-Guided Gradient Constraint is explored to dynamically control model updates according to changes in knowledge measured by the learned prototypes, achieving a better balance between preserving acquired knowledge and learning new information.

To sum up, the main contributions of this work are three-fold: (1) To tackle the challenging and critical NEOCL problem, we propose a novel Progressive Prototype Evolving method that effectively learns prototypes and guides the model to mitigate both intra and

inter-stage forgetting. (2) The prototypes are progressively learned and concurrently fed back to the current learning stage to mitigate intra-stage forgetting. (3) Moreover, Prototype Similarity Preserving and Prototype-Guided Gradient Constraint are introduced to counter inter-stage forgetting, striking a balanced approach between knowledge retention and acquisition.

## 2 RELATED WORK

### 2.1 Continual Learning

Current Continual Learning methods can be broadly categorized into *rehearsal-based*, *regularization-based*, and *architecture-based* models. Among them, rehearsal-based approaches [27, 28, 34] concentrated on preserving knowledge by actively selecting and replaying representative data from earlier classes. Moreover, regularization-based methods [21, 24, 39, 40] aimed to address forgetting by stabilizing model parameters or regulating feature adjustments. The architecture-based CL models [18, 43, 44, 50] adapted dynamically to evolving data streams by either modifying network structures or incorporating specific parameters tailored for each learning stage. However, the aforementioned methods usually have to store historical data, raising critical concerns about data privacy.

Consequently, recent efforts have been made to address CL without retaining historical exemplars, leading to Non-Exemplar Continual Learning. In this context, the challenge of catastrophic forgetting intensifies due to the absence of previous data. To address this, [10, 11, 24] proposed knowledge distillation to resist forgetting. [51, 52] emphasized data augmentation to broaden the classification boundary and diminish the representation bias in continual learning. Several methods maintained prototypes of each class as surrogates for prior knowledge. Most of them [31, 33, 37, 53] computed prototypes as the mean feature of each class after the training process and proposed various prototype augmentation and reminiscence techniques to retain past knowledge. The latest research [3] treated prototypes as learnable parameters during the training process. However, these methods typically utilized previously learned prototypes solely to mitigate knowledge forgetting in the following learning stages, without fully harnessing the information from the prototypes of the current stage.

### 2.2 Online Continual Learning

Recently, a more challenging but realistic CL scenario, named Online Continual Learning, has been investigated, where each sample can only be accessed once. Existing OCL methods suffered from not only inter-stage forgetting when learning unseen classes but also intra-stage forgetting of previously learned knowledge within a learning stage. Therefore, various OCL-specific approaches have emerged that predominantly rely on retaining previous exemplars as the rehearsal. [2, 20] focused on storing balanced and representative exemplars from the learned stages, and [1, 38, 41] proposed to select valuable samples for replay, based on the gradient or class information. In addition to preserving exemplars, several approaches [4, 47] concentrated on enhancing the learning capability of models by either incorporating supervised contrastive learning [26, 30, 46] or employing mutual information [13, 15] to facilitate OCL. Considering that it is infeasible to directly compute the average features of all samples as prototypes in OCL, besides

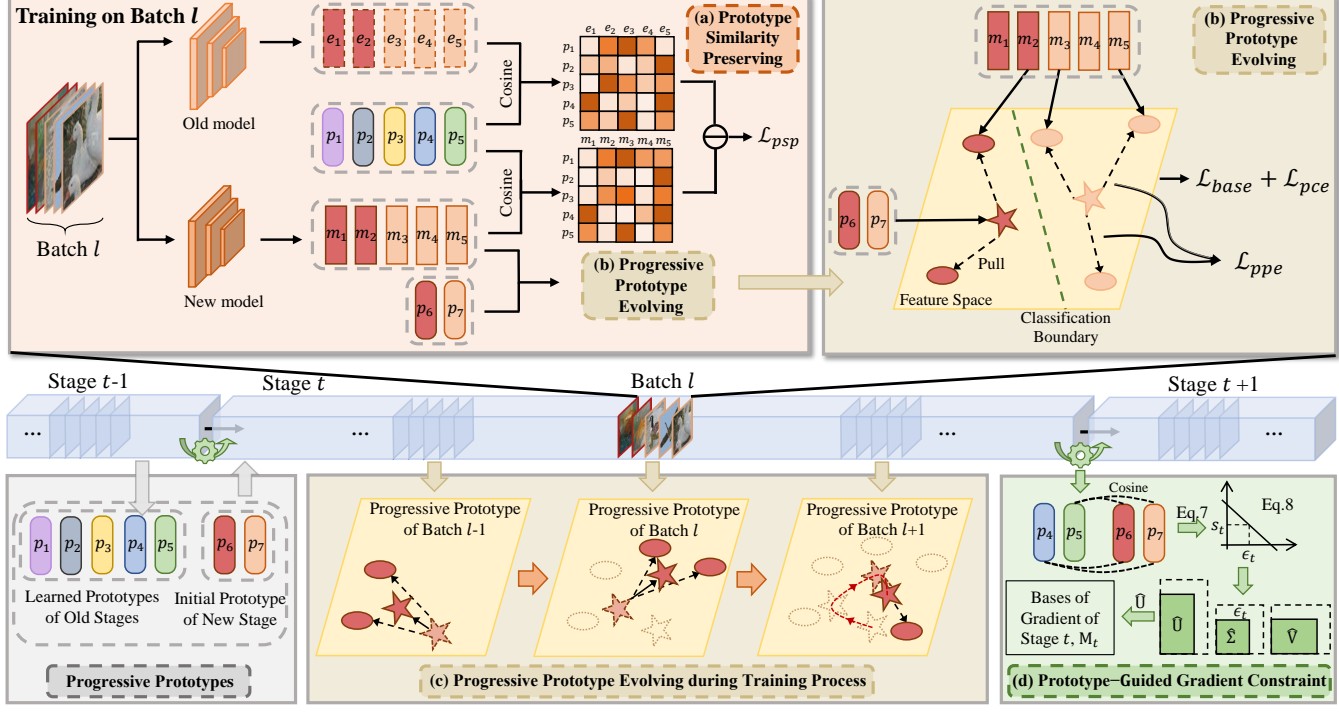

**Figure 3: The overall pipeline of our proposed PPE. (a) During the training of the $l$-th batch in the $t$-th learning stage, a Prototype Similarity Preserving mechanism is introduced to mitigate inter-stage forgetting. (b) Meanwhile, the designed learnable prototypes progressively evolve through the optimization of the model and are readily integrated into the training of the current stage to alleviate intra-stage forgetting. (c) shows the overall progressive evolution of prototypes between adjacent batches. (d) Finally, to preserve more previously learned information, the Prototype-Guided Gradient Constraint is proposed to dynamically restrict model updates according to the changes in knowledge measured by the learned prototypes after each learning stage.**

keeping exemplars, CoPE [7] also maintained prototypes as extra knowledge through momentum updating, but the obtained prototypes might be misled by the latest samples.

The aforementioned methods all need to preserve historical exemplars, which obviously hinders data privacy and contradicts the online setting. In this paper, we focus on tackling OCL without preserving any exemplars, in which rare works have been proposed. To preserve knowledge without assessing to previous data, [8] employed knowledge distillation for old classes but it ignored the intra-stage forgetting. [14] concentrated on adopting orthogonal projection to mitigate forgetting, but its learning capability is severely restricted, resulting in unsatisfactory performance. The latest approach [19] focused on learning a pre-trained model in an offline manner on the base stage and subsequently froze the pre-trained backbone during online learning. Though the forgetting problem can be addressed with the frozen backbone, the learning ability is limited especially when the training data of the base stage is insufficient. In contrast, our method entails continual learning from scratch without relying on an offline training stage. The proposed PPE model leverages a progressive evolving strategy to learn informative and discriminative prototypes that convey helpful knowledge to mitigate intra and inter-stage forgetting issues.

## 3 THE PROPOSED METHOD

### 3.1 Problem Formulation

NEOCL considers learning a model continually from a single-pass stream of $T$ stages $\mathcal{D} = \{D_1, D_2, ..., D_T\}$ without storing any exemplars of previous data. The model $\Theta = \{\Phi, \Psi\}$ consists of a feature extractor $\Phi$ and a classifier $\Psi$. Data of the $t$-th stage $D_t = \{X_t, Y_t\}$ consists of an image set $X_t = \{x_{t,i}\}_{i=1}^{n_t}$ and a class label set $Y_t = \{y_{t,i} \in C_t\}_{i=1}^{n_t}$, where $n_t$ is the number of data in stage $t$ and $C_t$ represents the set of classes in stage $t$. Notably, each input sample $x_{t,i}$ can only be used once, and labels of different stages are disjoint.

### 3.2 Mitigate Intra-stage Forgetting

**Progressive Prototype Evolving.** Firstly, we introduce the acquisition of our progressively learned prototypes to further demonstrate the collaboration between prototype evolution and intra-stage forgetting mitigation. Since no previous data can be accessed in NEOCL, we propose to handle catastrophic forgetting by employing learnable prototypes as the representative of essential knowledge for each class. Specifically, the prototypes are treated as a set of learnable parameters and supervised with the features of respective classes. At the beginning of the $t$-th learning stage, we

initialize a class-specific prototype for each class of the current stage randomly $P_t = \{p_i | i \in C_t\}$. Then given a batch of data, we optimize the prototypes by minimizing the distance between the features and the prototypes as follows:

$$\mathcal{L}_{ppe} = \frac{1}{n_b} \sum_{i=1}^{n_b} \|\text{stop}(\Phi(x_i)) - p_{y_i}\|_2^2, \tag{1}$$

where $n_b$ represents the batch size, $p_{y_i}$ denotes the prototype for class $y_i$, and stop refers to the operation that stops the gradient backward. Through this optimization process, prototypes in $P_t$ can progressively acquire class-specific information within this batch. As a result, the prototypes $P_t$ progressively evolve over the whole learning process instead of solely relying on the latest batch, they effectively accumulate knowledge of previously encountered samples within a stage.

To mitigate the critical intra-stage forgetting issue in OCL and fully leverage the knowledge compressed in $P_t$, we incorporate prototypes into the model training process of the $t$-th learning stage. Specifically, $P_t$ are fed to the classifier:

$$\mathcal{L}_{pce} = \mathcal{L}_{\text{CE}}(\Psi(P_t), Y_{p,t}), \tag{2}$$

where $\mathcal{L}_{\text{CE}}$ and $Y_{p,t}$ denote the cross-entropy loss and labels associated with prototypes $P_t$. Through this operation, the accumulated knowledge within $P_t$ is readily replayed to mitigate intra-stage forgetting and enhance the training of a more robust model. Consequently, the learning of the model and the progressive evolution of prototypes are mutually reinforcing. A more robust model contributes to the evolution of prototypes, resulting in more informative prototypes.

## 3.3 Mitigate Inter-stage Forgetting

**Prototype Similarity Preserving.** To alleviate the commonly mentioned inter-stage forgetting problem, existing NECL methods [37, 52] always apply a hard loss function that minimizes the Euclidean distance between the features of input data extracted by the current and old models. However, such a paradigm severely limits the learning capability of the model and leads to unsatisfied performance in NEOCL, where the model suffers from insufficient training. Thus, to mitigate inter-stage forgetting while maintaining the plasticity of knowledge acquisition, we propose to distill the knowledge of similarity between prototypes and input data. Specifically, following [22] which shows that a model updated with model fusion can better represent the accumulated knowledge, we maintain a fusion model denoted as $\Theta_e = \{\Phi_e, \Psi_e\}$.

Then similarity between the prototypes and the features of input data, extracted by the current model $\Theta$ and the old model $\Theta_e$, is calculated via cosine similarity. The proposed Prototype Similarity Preserving loss can be formed as:

$$\mathcal{L}_{psp} = \frac{1}{n_b \cdot |C_t|} \sum_{i=1}^{n_b} \sum_{j \in C_t} \|\langle \Phi_e(x_i), p_j \rangle - \langle \Phi(x_i), p_j \rangle\|_1, \tag{3}$$

where $\langle \cdot, \cdot \rangle$ represents the cosine similarity, $n_b$ represents the batch size, and $C_t = \bigcup_{i=1}^{t-1} C_i$ represents labels of classes that have been seen during previous stages. By employing $\mathcal{L}_{psp}$, the semantic relationships between data features and prototypes can be distilled to resist inter-stage forgetting.

**Prototype-Guided Gradient Constraint.** To further preserve more informative knowledge of historical stages, motivated by [36], we design a Prototype-Guided Gradient Constraint scheme to mitigate the inter-stage forgetting by orthogonal gradient projection. When training on the $l$-th batch of the $t$-th stage, we remove the projection of the gradient $g_{t,l}$ into the space of gradients from previous stages to mitigate its influence on these stages:

$$\hat{g}_{t,l} = g_{t,l} - M_{t-1}M_{t-1}^T g_{t,l}, \tag{4}$$

where $M_{t-1} = [u_1, u_2, ...]$ represents the orthogonal bases of subspace which contain the gradients of previous stages. To obtain $M_t$, we leverage the proposition that the gradient update of each layer lies in the span of inputs as illustrated in Supplementary. After the training process of each stage, an input matrix $R_t$ of each layer is constructed. The part of $R_t$ that can be represented by $M_{t-1}$ is removed: $\hat{R}_t = R_t - M_{t-1}M_{t-1}^T R_t$. Then the SVD operation is performed on $\hat{R}_t = \hat{U}_t \hat{\Sigma}_t \hat{V}_t$ where $\hat{U}_t = [\hat{u}_{t,1}, \hat{u}_{t,2}, ...]$ and a minimum value of top-$k$ rank approximation $(\hat{R}_t)_k$ is chosen to meet the following criteria given a hyperparameter $\epsilon_t$:

$$\|(\hat{R}_t)_k\|_F^2 + \|M_{t-1}M_{t-1}^T R_t\|_F^2 \geq \epsilon_t \|R_t\|_F^2. \tag{5}$$

Then $M_t$ is updated as $[M_{t-1}, \hat{u}_{t,1}, ..., \hat{u}_{t,k}]$. Existing methods [25, 36] construct $R_t$ via sampling data from the entire dataset which is infeasible in NEOCL. Therefore, we handle this constraint by utilizing the latest batch of a learning stage to obtain $R_t$.

In fact, the knowledge of different periods of a data stream in NEOCL is always unpredictable and may lead to varying degrees of model parameter drifting during learning. Therefore, instead of relying on a high threshold $\epsilon_t$ to mitigate forgetting, we argue that an adaptive threshold parameter is essential to guide the update of orthogonal bases effectively. To do so, we leverage the obtained progressive prototypes to measure the newly acquired knowledge of the current stage and guide the update of orthogonal bases. Specifically, after the training process of the $t$-th stage, we calculate the average cosine similarity between the set of prototypes for classes in stages $t$-1 and $t$ as:

$$s_t = \frac{1}{|C_{t-1}| \cdot |C_t|} \sum_{i \in C_{t-1}} \sum_{j \in C_t} \langle p_i, p_j \rangle. \tag{6}$$

A lower average similarity implies that the features of new classes are significantly different from those of previously seen classes, indicating that the model has learned more novel knowledge during this stage. Consequently, a larger $\epsilon_t$ is assigned to preserve more information about the current stage. Thus, we choose a linear function with negative scope to model this *negative correlation* between the $\epsilon_t$ in Equation (5) and $s_t$ as:

$$\epsilon_t = \alpha - \beta s_t, \tag{7}$$

where $\alpha \geq 0, \beta \geq 0$ are hyperparameters.

## 3.4 Overall Optimization

The overall pipeline of our proposed PPE method is shown in Figure 3. For optimization, following [26, 30], a base loss $\mathcal{L}_{base}$ consisting of a cross-entropy loss and a supervised contrastive learning is adopted to train the model. The proposed $\mathcal{L}_{ppe}$ in Equation (1) is incorporated for progressive prototype evolving, and then the learned prototypes are fed to the classifier based on the prototype

**Table 1: The Final Accuracy (higher is better) of different methods on various datasets. All experiment results are the average performance across 15 runs. The backbone of all comparison methods is the reduced ResNet18 except AOP marked by * which uses the AlexNet.**

| Data & Memory Size | | | CIFAR-10 | | | CIFAR-100 | | | MiniImageNet | | |
|---|---|---|---|---|---|---|---|---|---|---|---|
| | | | 10 | 20 | 100 | 100 | 200 | 500 | 100 | 200 | 500 |
| Exemplar-Based | ER [6] | arXiv2019 | 18.2±0.2 | 18.6±0.4 | 21.0±0.7 | 7.5±0.2 | 7.9±0.3 | 9.6±0.5 | 5.6±0.2 | 6.2±0.2 | 7.6±0.4 |
| | MIR [1] | NeurIPS2019 | 18.3±0.3 | 18.6±0.4 | 20.7±0.4 | 7.4±0.2 | 7.9±0.2 | 9.5±0.3 | 5.8±0.2 | 6.1±0.2 | 7.7±0.4 |
| | CoPE [7] | ICCV2021 | 22.1±1.1 | 24.5±1.2 | 31.6±1.8 | 7.0±0.2 | 7.5±0.2 | 8.9±0.3 | 3.9±0.2 | 4.5±0.2 | 5.8±0.3 |
| | SCR [30] | CVPR-W2021 | 20.0±2.1 | 26.2±2.2 | 39.3±1.3 | 9.2±0.3 | 12.7±0.4 | 19.8±0.4 | 7.9±0.3 | 10.9±0.3 | 17.8±0.4 |
| | OCM [15] | ICML2022 | 25.6±2.5 | 27.5±1.2 | 45.6±1.2 | 6.3±0.4 | 9.8±0.4 | 17.0±0.6 | 4.3±0.2 | 7.0±0.4 | 12.6±0.5 |
| | RAR [48] | NeurIPS2022 | 18.9±1.5 | 23.6±1.5 | 36.0±1.1 | 9.7±0.4 | 13.3±0.3 | 19.0±0.4 | 8.7±0.4 | 12.1±0.4 | 18.2±0.3 |
| | DVC [13] | CVPR2022 | 22.1±1.7 | 26.4±0.2 | 40.0±1.9 | 10.4±0.4 | 12.6±0.5 | 16.0±0.8 | 8.5±0.4 | 9.6±0.7 | 13.0±0.8 |
| | GSA [16] | CVPR2023 | 25.5±1.3 | 30.3±0.9 | 47.1±1.1 | 11.8±0.3 | 14.5±0.5 | 20.4±0.3 | 9.6±0.3 | 12.1±0.3 | 16.7±0.3 |
| | PCR [26] | CVPR2023 | 26.7±1.8 | 33.4±1.4 | 45.9±1.5 | 13.4±0.6 | 16.8±0.3 | 21.8±0.6 | 12.6±0.5 | 15.3±0.8 | **19.6±0.8** |
| | CBA [45] | ICCV2023 | 21.8±1.6 | 25.0±1.8 | 39.6±1.3 | 11.8±0.5 | 15.2±0.7 | 19.8±1.1 | 7.6±0.5 | 9.8±0.6 | 12.3±0.8 |
| | OnPro [46] | ICCV2023 | 23.0±2.8 | 31.7±1.9 | **50.1±1.7** | 8.3±0.4 | 10.9±0.5 | 16.5±0.5 | 5.1±0.3 | 7.5±0.4 | 11.7±0.4 |
| | SSD [12] | AAAI2024 | 20.7±0.8 | 22.3±0.7 | 38.7±1.1 | 9.5±0.3 | 13.4±0.4 | 21.9±0.4 | 8.7±0.3 | 13.2±0.4 | 19.1±0.5 |
| Non-Exemplar | Memory size | | 0 | 0 | 0 | 0 | 0 | 0 | 0 | 0 | 0 |
| | PASS [52] | CVPR2021 | 28.5±0.9 | 28.5±0.9 | 28.5±0.9 | 8.0±0.9 | 8.0±0.9 | 8.0±0.9 | 2.4±0.6 | 2.4±0.6 | 2.4±0.6 |
| | AOP* [14] | AAAI2022 | 42.7±0.6 | 42.7±0.6 | 42.7±0.6 | 11.2±0.3 | 11.2±0.3 | 11.2±0.3 | 7.4±0.2 | 7.4±0.2 | 7.4±0.2 |
| | PRAKA [37] | ICCV2023 | 33.9±1.7 | 33.9±1.7 | 33.9±1.7 | 5.9±0.8 | 5.9±0.8 | 5.9±0.8 | 2.4±0.5 | 2.4±0.5 | 2.4±0.5 |
| | DSR [19] | AAAI2024 | 22.4±0.2 | 22.4±0.2 | 22.4±0.2 | 6.4±0.1 | 6.4±0.1 | 6.4±0.1 | 5.1±0.1 | 5.1±0.1 | 5.1±0.1 |
| | **PPE(Ours)** | **This Paper** | **43.2±0.6** | **43.2±0.6** | 43.2±0.6 | **22.0±0.4** | **22.0±0.4** | 22.0±0.4 | **16.9±0.5** | **16.9±0.5** | 16.9±0.5 |

classification loss $\mathcal{L}_{pce}$ to mitigate the intra-stage forgetting. Moreover, a prototype similarity preserving loss $\mathcal{L}_{psp}$ is also involved to mitigate the problem of inter-stage forgetting. Finally, the overall loss function in the proposed method can be formed with weight parameters $\lambda, \gamma, \mu$:

$$\mathcal{L} = \mathcal{L}_{base} + \lambda\mathcal{L}_{ppe} + \gamma\mathcal{L}_{pce} + \mu\mathcal{L}_{psp}. \tag{8}$$

To be noted, the gradient of the model is constrained by the Prototype-Guided Gradient Constraint. At the end of each learning stage, the orthogonal bases are updated with the adaptive threshold in Equation (7).

### 3.5 Discussion

Maintaining prototypes is a kind of rising method in the area of NECL and OCL. In this section, we aim to elucidate the distinctions between our PPE method and existing techniques. (1). We propose *progressive prototype* that treat the prototypes as learnable parameters and optimize them during training. This prototype fits the online learning scenarios where conventional methods, such as computing the mean feature of samples from the same class, are impractical. (2). To *mitigate intra-stage forgetting*, we propose the $\mathcal{L}_{pce}$ which incorporates the accumulated knowledge obtained by our evolving prototypes. As discussed in Section 1, apart from the well-known inter-stage forgetting, we demonstrate that NEOCL also meets with the intra-stage forgetting that the model may forget the knowledge of previously seen samples during a learning stage. The effectiveness of our proposed $\mathcal{L}_{pce}$ is verified in Section4.3 and Supplementary. (3). The knowledge of prototypes is fully explored to *mitigate inter-stage forgetting*. Firstly, drawing inspiration

from knowledge distillation techniques, we introduce $\mathcal{L}_{psp}$ to distill knowledge using our progressive prototypes. Secondly, to address the limited learning capability of the Gradient Projection Memory (GPM) methods and strike a balance between knowledge acquisition and forgetting, we design an adaptive threshold for GPM based on the estimated knowledge shift between different stages, as indicated by the similarity of prototypes.

## 4 EXPERIMENTS

### 4.1 Experiment Settings

**Datasets.** The evaluation of our method is conducted on three widely used datasets, CIFAR-10 [23], CIFAR-100 [23], and MiniImageNet [42]. We follow previous works [26, 30] to build the OCL setting. Specifically, CIFAR-10 is divided into 5 stages, each comprising 2 classes, while CIFAR-100 and MiniImageNet are divided into 10 stages, each encompassing 10 classes.

**Comparison Methods.** We compare our PPE with various methods including twelve OCL models (ER [6], MIR [1], CoPE [7], SCR [30], OCM [15], RAR [48], DVC [13], GSA [16], PCR [26], CBA [45], On-Pro [46], SSD [12]) and four NECL approaches (PASS [52], AOP [14], PRAKA [37], DSR [19]). For OCL methods that need to store and replay data, we set the memory size with 10/20/100 for CIFAR-10, as well as 100/200/500 for CIFAR-100 and MiniImageNet which are widely adopted by OCL methods [1]. To be noted, NECL methods and PPE do not need to store exemplars, thus the memory size is 0.

---

[1]For example, a memory size of 100 for CIFAR-10 means keeping 10 exemplars per class.

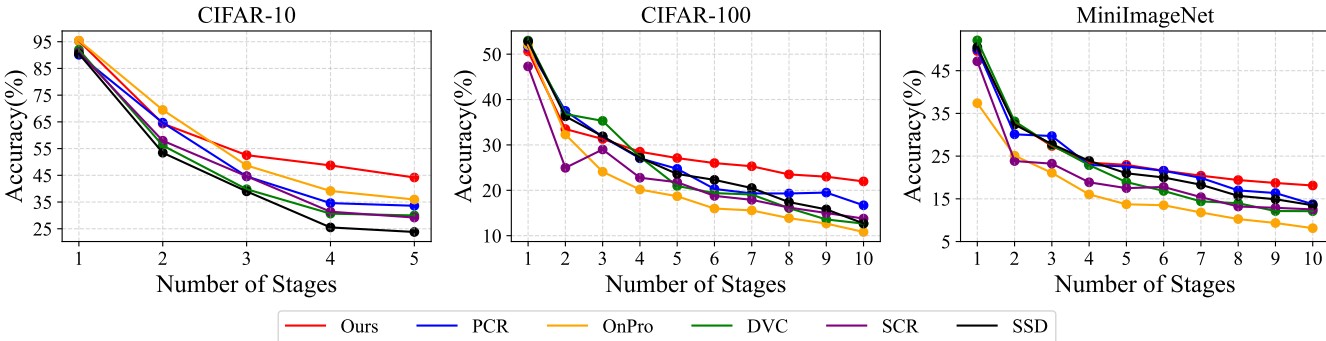

Figure 4: The complete classification accuracy of different methods on each stage.

Table 2: The Average Forgetting (lower is better) of different methods on various datasets.

| Methods | CIFAR-10 | CIFAR-100 | MiniImageNet |
|---|---|---|---|
| SCR [30] | 56.9±2.0 | 26.7±0.8 | 17.8±1.0 |
| RAR [48] | 61.3±3.0 | 41.9±0.8 | 24.9±1.1 |
| DVC [13] | 48.1±2.8 | 39.0±0.9 | 34.2±0.9 |
| GSA [16] | 57.3±1.4 | 48.4±0.6 | 38.8±0.6 |
| PCR [26] | 37.0±4.2 | 31.3±0.8 | 30.5±0.9 |
| SSD [12] | 64.9±1.0 | 32.7±0.7 | 18.0±0.8 |
| **PPE(Ours)** | **23.3±0.7** | **14.8±0.6** | **10.8±0.5** |

**Evaluation Metrics.** Following [13, 38, 46], the *Final Accuracy* and *Average Forgetting* [5] are adopted for evaluation. Final Accuracy is computed as the accuracy of all seen classes and Average Forgetting calculates the average accuracy degradation of different classes, which represents the anti-forgetting ability of the model.

**Implementation Details.** For a fair comparison, the reduced ResNet18 [17] trained from scratch is used as the backbone. Following [16, 46], we use resized-crop, random flip, and gray-scale as data augmentation strategies. At each training stage, the classification head related to the current stage are optimized. During testing, prototypes are treated as class means and the nearest class mean classifier [30, 32] is used. For hyperparameters, in Equation (7), $\alpha = 2.65$, $\beta = 2$. The weighting parameters of different losses are $\gamma = 0.5$, $\lambda = 1$, and $\mu = 10$. Optimization of model parameters and progressive prototypes are performed using the SGD optimizer with learning rates of 0.02 and 60 respectively. Experiments of hyperparameters are shown in the Supplementary. The batch size is 10 for all methods. For the comparison methods, we reproduce their results with their official source code and all methods use the reduced ResNet18 backbone except for AOP which uses AlexNet. All results are the average performance across 15 runs.

## 4.2 Comparison with SOTA

**Comparison of Final Accuracy.** Table 1 reports the Final Accuracy of the comparison methods. Across various scenarios, without storing any exemplars, our PPE significantly outperforms existing OCL methods and achieves comparable results when they preserve

a large number of exemplars. Specifically, the performance of existing OCL methods will dramatically degrade with limited exemplars. For example, on the CIFAR-100 dataset, the performance of the latest method SSD drops from 21.9% to 9.5% when the memory size goes smaller while our PPE achieves superior results of 22.0% without reusing any previous exemplars. Compared with NECL methods, our PPE outperforms them by 0.5%, 10.8%, and 9.5% on CIFAR-10, CIFAR-100, and MiniImageNet. Specifically, although AOP utilizes the AlexNet which is more suitable for CIFAR-10, our method can still beat it. This can be attributed to the comprehensive utilization of our progressively evolved prototypes. Different from the prototypes in these NECL methods which are solely used to resist the inter-stage forgetting of classification heads, our prototypes are able to jointly mitigate the intra and inter-stage forgetting.

**Comparison of Average Forgetting.** In Table 2, we present the Average Forgetting results of the state-of-the-art exemplar-based OCL methods that achieve promising Final Accuracy results. The memory sizes are 20/200/200 for CIFAR-10 /CIFAR-100 /MiniImageNet. It is obvious that, despite the extensive use of exemplars in previous methods, they exhibit higher forgetting rates compared to our PPE. Since the existing OCL methods primarily rely on reusing abundant exemplars without designing specific components to mitigate forgetting. Our approach, on the one hand, introduces Progressive Prototype Evolving and Prototype Similarity Preserving loss to address the dual-forgetting, and the proposed Prototype-Guided Gradient Constraint aids in achieving a more balanced trade-off between knowledge retention and acquisition.

**Accuracy of Different Stages.** To present results in detail, the classification accuracy of different methods on sequential learning stages are shown in Figure 4. The memory sizes are 20/200/200 for CIFAR-10/CIFAR-100/MiniImageNet. Notably, during the early stages, certain exemplar-based OCL methods achieve higher accuracy than our approach. This is because these methods leverage a fixed-size memory buffer, resulting in more exemplars per class in the early stages, thereby enhancing the performance. However, as training advances, the stored exemplars per class continue to decrease which exacerbates the problem of catastrophic forgetting causing their accuracy to decline. Eventually, our proposed PPE method outperforms these counterparts, yielding superior Final Accuracy results in the long run.

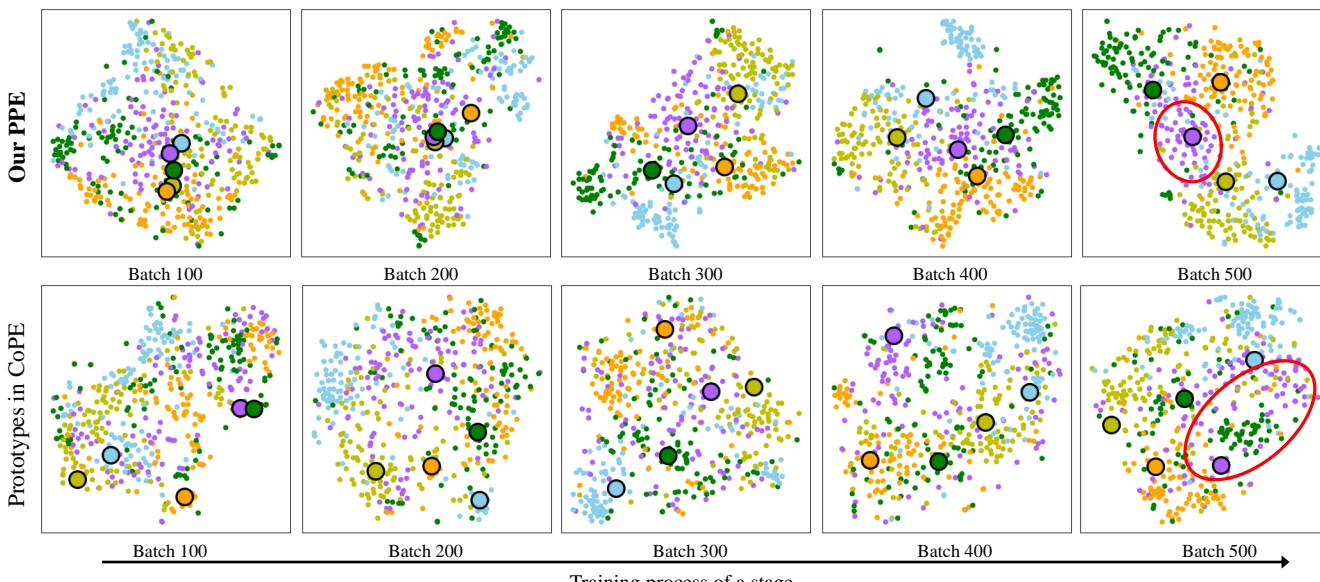

**Figure 5: The t-SNE visualization results of prototypes and extracted features on test sets at different batches on CIFAR-100. Data features are represented by small circles and prototypes are represented by big circles with black edges.**

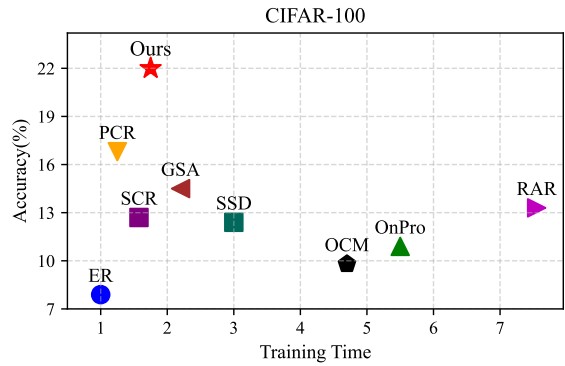

**Figure 6: The mean training time of different methods. All training time results are normalized to ER.**

**Comparison of Training Time.** In Figure 6, we conduct a comparative analysis of training times among various methods using CIFAR100 with a memory size of 200. As can be observed, our PPE demonstrates superior performance, although the training time is slightly higher than ER due to the gradient projection operation introduced in Section 3.3. Notably, our approach significantly outperforms RAR, OCM, and OnPro, which introduce repeated sampling from memory buffers or employ strong data augmentation techniques. This efficiency advantage makes our approach highly adaptable to the demands of the online scenario.

## 4.3 Ablation Study

**Effectiveness of Different Components.** Ablation results are presented in Table 3. Our method incorporates four losses $\mathcal{L}_{base}$,

**Table 3: Ablation study of different components on CIFAR-100 and MiniImageNet datasets.**

| $\mathcal{L}_{psp}$ | $\mathcal{L}_{pce}$ | PGC | **CIFAR-100** | **MiniImageNet** |
|---|---|---|---|---|
| - | - | - | 19.1±0.3 | 14.5±0.3 |
| ✓ | - | - | 19.6±0.3 | 14.9±0.3 |
| ✓ | ✓ | - | 20.2±0.3 | 15.6±0.5 |
| ✓ | ✓ | ✓ | **22.0±0.4** | **16.9±0.5** |

**Table 4: Gradient constraint with different fixed thresholds on CIFAR-100 dataset.**

| $\epsilon_t$ | 0.3 | 0.5 | 0.7 | 0.75 |
|---|---|---|---|---|
| Acc(%) | 13.1±0.3 | 17.0±0.3 | 21.2±0.3 | 21.6±0.3 |
| $\epsilon_t$ | 0.8 | 0.85 | 0.9 | **PPE(Ours)** |
| Acc(%) | 21.6±0.3 | 21.5±0.4 | 20.2±0.3 | **22.0±0.4** |

$\mathcal{L}_{ppe}$, $\mathcal{L}_{pce}$, $\mathcal{L}_{psp}$ and Prototype-Guided Gradient Constraint. Notably, due to the stop gradient operation in $\mathcal{L}_{ppe}$, the learning of prototypes does not impact the backbone when only $\mathcal{L}_{base}$ is employed. Thus the model optimized by $\mathcal{L}_{base} + \delta\mathcal{L}_{ppe}$ and updated by gradient projection with fixed threshold, 0.9, is selected as baseline. Table 3 shows that the usage of $\mathcal{L}_{psp}$ leads to an improvement of 0.5%/0.4%, underscoring the effectiveness of Prototype Similarity Preservation in mitigating inter-stage forgetting. Additionally, $\mathcal{L}_{pce}$ further boosts the results by 0.6%/0.7% which can be attributed to the well-handling of intra-stage forgetting via our proposed Progressive Prototype Evolving. Finally, the full model integrating Prototype-Guided Gradient Constraint achieves the best results, validating the prototypes' effect of guiding the model update.

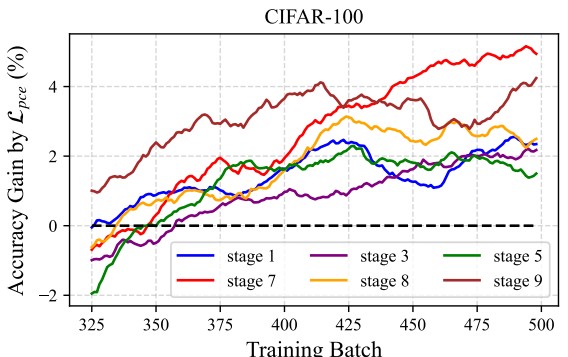

Figure 7: Accuracy gain of previously seen samples brought by $\mathcal{L}_{pce}$.

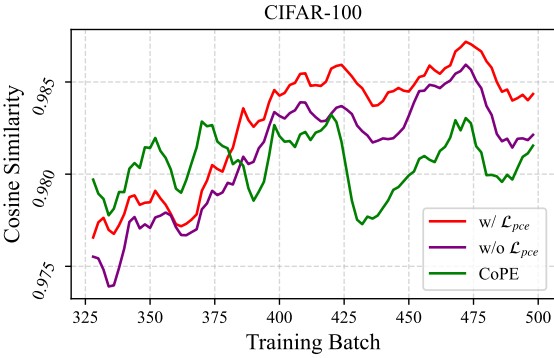

Figure 8: The average cosine similarity between the learned prototypes and the calculated mean feature of test data.

To further illustrate the effectiveness of PGC, we present results adopting fixed thresholds $\epsilon_t$ on CIFAR-100 in Table 4. Observably, performance degrades with fixed high or low thresholds. This is because a high threshold imposes strict constraints, limiting knowledge acquisition, while a low threshold compromises anti-forgetting ability. In contrast, employing thresholds calculated by prototypes achieves the best results, emphasizing the crucial role of the prototype's knowledge in guiding the model to strike an appropriate balance between knowledge acquisition and forgetting.

**Visualization of Progressive Prototypes**. In Figure 5, we present the t-SNE visualization results of prototypes and extracted features of images in test sets at different batches. The results of different batches within an online continual learning stage show the evolution of our progressive prototypes alongside the prototypes obtained through the momentum updating method CoPE [7]. For a fair comparison, no memory buffer is used in this experiment. It can be observed that, as the training progresses, the features of distinct classes become increasingly distinguishable and the prototypes evolve along with features. Our learnable prototypes demonstrate a superior ability to approximate the center of each class's features, while the prototypes acquired through momentum updating in CoPE are significantly influenced by later encountered samples and deviate from the true class centers. Notably, the superior discriminative ability of feature representations learned by our method is mainly because of the exploration of the knowledge encoded in the progressive prototypes to mitigate intra-stage forgetting, enhancing the overall capability of the model.

**Intra-stage Forgetting Mitigation.** Intra-stage forgetting refers to the phenomenon that the model gradually loses knowledge of previously encountered samples belonging to the same class within a specific training stage. Thus, a direct method to illustrate intra-stage forgetting is testing the model's performance on previously seen training samples during subsequent training batches. For a training stage with 500 batches on CIFAR-100 datasets (a total of 10 classes with 5000 samples, 10 samples per batch), given the potential insufficient training in the initial batches, where the model may not sufficiently learn from the early samples, we opt to use samples from batches 250 to 300 for constructing the test set. In Figure 7 we demonstrate the effectiveness of our anti-intra-stage forgetting module, $\mathcal{L}_{pce}$, by showcasing the accuracy improvement

brought by $\mathcal{L}_{pce}$ on our constructed test set. The results reveal a consistent enhancement in accuracy with the incorporation of $\mathcal{L}_{pce}$. Furthermore, the magnitude of this improvement increases with the progression of training batches. This is because as the training progresses, forgetting becomes more severe. $\mathcal{L}_{pce}$ can effectively mitigate intra-stage forgetting by leveraging the accumulated knowledge acquired through our progressive prototypes, thereby contributing to the observed performance enhancement. More verification of intra-stage forgetting mitigation is provided in Supplementary.

**Collaboration between PPE and Intra-stage Forgetting Mitigation.** In this part, we delve into the synergistic relationship between our proposed PPE and the mitigation of intra-stage forgetting. To demonstrate the enhancement brought by $\mathcal{L}_{pce}$ to the learning process of prototypes, we compare the average cosine similarity between the class mean feature on test sets and prototypes at each batch in Figure 8. A higher similarity signifies closer alignment between learned prototypes and their appropriate positions. Compared with other prototype acquisition methods, the momentum updating-based CoPE [7] and solely learning prototypes without $\mathcal{L}_{pce}$, our method achieves superior similarity results indicating that the learned prototypes are closer to their appropriate position. This result demonstrates the positive impact of our mitigating intra-stage forgetting on prototype learning.

## 5 CONCLUSION

In this paper, we target the challenging yet crucial NEOCL problem and propose a novel method, named the Progressive Prototype Evolving, to tackle the dual-forgetting issues in NEOCL. To mitigate intra-stage forgetting, our approach learns class-specific progressive prototypes as surrogates for previous knowledge and leverages a prototype feedback design to utilize the accumulated knowledge of prototypes within a learning stage. Additionally, to resist inter-stage forgetting, our method incorporates Prototype Similarity Preserving and Prototype-Guided Gradient Constraint modules which explore prototypes to distill previous knowledge and constrain the model update. This work offers a fresh perspective on forgetting mitigation in OCL which highlights the knowledge of prototypes, getting rid of the reliance on exemplars.

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
