# OpenReview forum: "Progressive Prototype Evolving for Dual-Forgetting Mitigation in Non-Exemplar Online Continual Learning"
_acmmm.org/ACMMM/2024/Conference — MM2024 Poster_

### Official Review · Reviewer_bLwL · 2024-05-23

**Rating:** 4
**Confidence:** 3

**Summary:**

This paper proposes a non-exemplar online continual learning method that progressively learns class-specific prototypes. These prototypes as surrogates of previous knowledge of different classes can mitigate intra-stage forgetting when fed back to the model during training. Authors further propose the Prototype Similarity Preserving and Prototype-Guided Gradient Constraint so as to address the inter-stage forgetting.

**Strengths:**

This work researches the online continual learning problem which is more common than the traditional continual learning in real-world applications. The propose method achieves higher performance than existing OCL methods with relatively low memory size in various datasets.

**Limitations:**

1.	What is the “stage” defined in the online continual learning? How are the different stages divided?
2.	The proposed prototype-guided gradient constraint is inspired by an individual category of continual learning methods, namely gradient projection memory [1]. How does the gradient projection memory perform in the online setting?
3.	For the prototype-guided gradient constrain method, a memory unit $M_t$ is utilized to store bases of subspaces of previous stages. How much memory is consumed by this memory unit?
4.	How is the hyper-parameter $\lambda$ in Equation (8) chosen? Authors are encouraged to provide some results to show its influence.

[1] Gobinda Saha,Isha Garg,and Kaushik Roy. 2021. Gradient Projection Memory for Continual Learning. In International Conference on Learning Representations. https://openreview.net/forum?id=3AOj0RCNC2

**Suitability:**

2

---

### Official Review · Reviewer_Sonu · 2024-05-24

**Rating:** 4
**Confidence:** 4

**Summary:**

This paper introduces a novel approach to Online Continual Learning without saving exemplars. The proposed method, Progressive Prototype Evolving (PPE), progressively learns and evolves class-specific prototypes during the online learning phase to mitigate both intra-stage and inter-stage forgetting. The PPE approach leverages Prototype Similarity Preserving and Prototype-Guided Gradient Constraint modules to distill and leverage historical knowledge.

**Strengths:**

1. The prototype Similarity Preserving and Prototype-Guided Gradient Constraint modules are somewhat novel and technically sound to address the dual-forgetting problem effectively.
2. The paper is well-structured, with clear explanations, visual aids, and detailed experimental results, which enhance understanding.
3. The paper provides extensive experiments on three datasets (CIFAR-10, CIFAR-100, and MiniImageNet), showing that PPE outperforms these methods.

**Limitations:**

1. This work does not show the combined hyperparameter sensitivity analysis for the overall optimized target, especially considering there are three hyperparameters in equation 8. It is essential to prove robustness. In the supplementary file, you only show the influence of each one by fixing the values of the other two.
2. The idea of utilizing prototype similarity/relation is not novel enough [1,2]. It would be better if there were more discussions to differ yours from the existing ones.
[1] Multi-granularity knowledge distillation and prototype consistency regularization for class-incremental learning. Neural Networks, 2023.
[2] Prototype reminiscence and augmented asymmetric knowledge aggregation for non-exemplar class-incremental learning. ICCV, 2023.
3. In the evaluation, the memory size seems too small for memory-based methods, failing to show their advantages. Meanwhile, you only compared your method with one method (DSR) that is specifically designed for exemplar-free OCL, which is insufficient.
4. I think there should be more comparisons and discussions to show the novelty and effectiveness more deeply.

**Suitability:**

2

---

### Official Review · Reviewer_oA42 · 2024-05-25

**Rating:** 3
**Confidence:** 4

**Summary:**

This paper proposes a novel online continual learning (OCL) method called Progressive Prototype Evolving (PPE) to tackle the specific dual-forgetting problem. Different from existing OCL methods, PPE effectively addresses the forgetting issue without using any previous exemplars. Specifically, PPE mitigates intra-stage forgetting using the progressive prototype evolving module, as well as mitigates inter-stage forgetting using the Prototype Similarity Preserving module and Prototype-Guided Gradient Constraint module. Extensive experiments on three widely used datasets demonstrate the superiority of the proposed PPE against the state-of-the-art exemplar-based OCL approaches.

**Strengths:**

1. This paper proposes a non-exemplar method for online continual learning (OCL), and I like this idea. Since existing OCL methods are almost all based on the replay of old data, these methods are susceptible to restrictions on accessing old data.
2. The elaboration of the paper is clear and reasonable.
3. The related work does help readers to understand the essence of the idea and to follow the work easily.
4. The experimental results are well presented and prove that the model is indeed effective. Many baselines are used.

**Limitations:**

1. In my opinion, this paper's biggest issue is its contribution to the multimedia community. Specifically, the proposed framework consists of four different loss functions, each of which is not very innovative and lacks theoretical depth for a paper on machine learning theory research. As for multimedia applications, the image classification task the paper selected is too simple.

2. The concept of prototypes has been explored in the continual learning community [1][2][3], and there is existing work on prototypes. What distinguishes the prototypes in this paper?

3. Why not take the weights in the classifier as the prototypes? In your proposed PPE method, you denote and optimize the prototypes based on the weights of the classifier. We generally take the weights of the classifier as the prototypes.

4. Is the learning rate 0.02 for all baseline methods? Why not set the learning rate as 0.02? In most research works, the learning rate is usually set to 0.1.

5. Show me the performance of the PPE method on both new knowledge and old knowledge at each learning stage.

6. Prove with experimental results that the Progressive Prototype Evolving module addresses the issue of intra-stage forgetting.

[1] Asadi N, Davari M R, Mudur S, et al. Prototype-sample relation distillation: towards replay-free continual learning[C]//International Conference on Machine Learning. PMLR, 2023: 1093-1106. [2] Shi W, Ye M. Prototype Reminiscence and Augmented Asymmetric Knowledge Aggregation for Non-Exemplar Class-Incremental Learning[C]//Proceedings of the IEEE/CVF International Conference on Computer Vision. 2023: 1772-1781. [3] Zhu K, Cao Y, Zhai W, et al. Self-promoted prototype refinement for few-shot class-incremental learning[C]//Proceedings of the IEEE/CVF conference on computer vision and pattern recognition. 2021: 6801-6810.

The following content does not affect my evaluation: I am one of the reviewers for the CVPR2024 submission version of this paper.

**Suitability:**

2

---

### Official Review · Reviewer_VvJQ · 2024-05-27

**Rating:** 4
**Confidence:** 2

**Summary:**

The paper introduces a Progressive Prototype Evolving (PPE), to address the challenges of catastrophic forgetting in non-exemplar online continual learning (NEOCL). Previous methods often rely on storing exemplars from previous tasks, which raises privacy concerns and is impractical in settings where data can only be accessed once. The proposed PPE method does not store past data; instead, it learns class-specific prototypes during the online learning phase. These prototypes serve as accumulated knowledge, which is used to mitigate both intra-stage and inter-stage forgetting. The effectiveness of PPE is demonstrated through extensive experiments on three benchmark datasets, showing its superiority over state-of-the-art exemplar-based and other non-exemplar continual learning approaches.

**Strengths:**

The paper presents thorough experimental results, comparing the PPE method against multiple baselines and existing state-of-the-art methods across different settings and datasets. The method is also detailed and clear, with explanations of the mechanisms like Prototype Similarity Preserving and Prototype-Guided Gradient Constraint

**Limitations:**

- The method appears to be computationally intensive due to the additional steps of learning and updating prototypes. There is a lack of discussions on the impact on computational scalability.
- The method involves multiple hyperparameters (like the weighting parameters of different losses), and there is a lack of sensitivity analysis.
- There is a lack of limitation discussions in this paper.

**Suitability:**

3

---

### Meta-Review · Area_Chair_WTw8 · 2024-06-30

**Recommendation:** Accept (Poster)
**Confidence:** 4

**Metareview:**

This paper proposes a non-exemplar method for online continual learning. It involves progressively learning and evolving class-specific prototypes during the online learning phase to mitigate both intra-stage and inter-stage forgetting.

In the meantime, it leverages prototype similarity preservation and prototype-guided gradient constraint modules to distill and leverage historical knowledge. The novelty is ok and the experiments are basically complete.

Most reviewers support this paper, and I believe some of its merits can outweigh its disadvantages. Therefore, I also hold a positive stand for this paper.